# Unusual Presentation of Aortic Valve Infective Endocarditis in a Dog: Aorto-Cavitary Fistula, Tricuspid Valve Endocarditis, and Third-Degree Atrioventricular Block

**DOI:** 10.3390/ani11030690

**Published:** 2021-03-04

**Authors:** Giovanni Romito, Alessia Diana, Antonella Rigillo, Maria Morini, Mario Cipone

**Affiliations:** Department of Veterinary Medical Sciences, University of Bologna, Ozzano dell’Emilia, 40064 Bologna, Italy; giovanni.romito2@unibo.it (G.R.); antonella.rigillo2@unibo.it (A.R.); maria.morini@unibo.it (M.M.); mario.cipone@unibo.it (M.C.)

**Keywords:** echocardiography, valvular vegetation, steno-insufficiency, aortocardiac shunt, canine

## Abstract

**Simple Summary:**

In dogs, infective endocarditis represents a rare but clinically relevant disease that typically involves the aortic and/or mitral valve. Transthoracic echocardiography plays an essential role in the diagnosis, prognosis, and monitoring of such a condition. Typical echocardiographic signs of disease progression include left-sided cardiac dilatation and extension of aortic and/or mitral vegetative lesions. Nevertheless, unexpected complications can sometimes develop, especially in the case of the erosion of the periannular tissue caused by lytic enzymes produced by bacteria. This case report describes the coexistence of multiple, uncommon complications of aortic valve infective endocarditis in a dog, namely, aorto-cavitary fistula, tricuspid valve endocarditis and third-degree atrioventricular block. In this study, the combination of the ante mortem (clinical and echocardiographic) and post mortem (gross pathology and histopathology) findings allowed us to gain detailed information on the disease process, its atypical complications, and the associated emodynamic consequences.

**Abstract:**

A 2-year-old Boxer with a history of subaortic stenosis and immunosuppressant therapy developed aortic valve infective endocarditis. On echocardiographic examination with simultaneous electrocardiographic tracing, multiple uncommon periannular complications of the aortic valve endocarditis were found, including aorto-cavitary fistula with diastolic left-to-right shunt, tricuspid valve endocarditis, and third-degree atrioventricular block. Necropsy confirmed the above echocardiographic findings. Although aortic valve endocarditis represents a well-known disease entity in dogs, the dynamic nature of this condition may allow development of complex and uncommon echocardiographic features.

## 1. Introduction

Infective endocarditis (IE) is defined as an inflammation of the endocardial surface of the heart from invasion by an infectious agent, usually bacterial [1,2]. Physiologically, host defense mechanisms adequately protect the endothelium from infections. Accordingly, the development of IE requires the simultaneous occurrence of several predisposing factors [1,2,3]. First, bacteremia is necessary for endocardial infection to occur [1,2]. In small animal practice, common sources of infections include dental procedures/periodontal disease, pyoderma, diskospondylitis, prostatitis, pneumonia, urinary tract infection, and long-term intravenous or urinary catheterization, although a definitive source of bacteremia is not identified in all cases [1,2,4]. The likelihood of IE becoming established is further increased when immunodeficiency disorders coexist or immunosuppressant therapies are prescribed [1,2,3]. Disrupted blood flow and endothelial damage represent two other important prerequisites for IE [1,2,3,5,6]. Endothelial damage usually arises from high velocity jets of blood passing through narrow orifices during systole, e.g., in the case of valvular stenosis. The interplay between hydrodynamics and endocarditis becomes particularly evident on the left-sided cardiac valves, naturally subjected to high strain due to the high pressures generated from the left ventricle [5,6]. Indeed, bacteria may be more easily forced into the endothelium of those valves experiencing higher resting pressures, and these valves may also be at increased risk for scarring [5,6]. This can explain why canine IE involves almost systematically the aortic and/or mitral valve and why subaortic stenosis (SAS) represents a perfect substrate for the development of this condition [1,2,5,6,7,8]. The effect of the mechanical trauma is to detach endothelial cells from its surface, exposing the subendothelial matrix, triggering a local coagulation response. Therefore, an aggregate of fibrinogen, fibrin, and platelet proteins develops, to which bacteria strongly bind [1,2,3]. Then, bacteria induce tissue factor production and platelet aggregation, thereby building a vegetative lesion [1,2,3]. Expansion of such vegetation provides bacteria with a formidable shield from the host defenses and antibiotic penetration [1,2,3]. At the same time, vegetations cause valve deformity and insufficiency/steno-insufficiency. Given the above, irregular-shaped masses adherent to the aortic and/or mitral valve represent the typical echocardiographic presentation of IE [1,2,9]. Nevertheless, sometimes the relentless progression of the infective process can transform classic echocardiographic findings into an atypical picture. Consequently, sonographers may face unexpected echocardiographic images. In this setting, knowledge of pathophysiology can significantly improve the echocardiographic interpretation, thus allowing sonographers to recognize even complex and atypical evolutions of IE. This case report describes an unusual presentation of aortic valve IE in a dog characterized by aorto-cavitary fistula (ACF), tricuspid valve endocarditis, and third-degree atrioventricular block (AVB).

## 2. Materials, Methods, and Results

### 2.1. Case Description and Clinical Investigations

A 32 kg, 2-year-old neutered male Boxer was referred to the Veterinary Teaching Hospital of the University of Bologna to evaluate a recent change in a previously diagnosed heart murmur associated with hyperthermia and lethargy. The dog was known to have a mild (2/6) left basilar systolic heart murmur due to a previously diagnosed mild SAS. Over time, multiple echocardiographic controls had been performed by the referring veterinarian (the last approximately one week prior to referral to our institution), showing neither progression of SAS nor development of additional cardiovascular abnormalities. Five days before referral, the dog developed lethargy and fever. A presumptive diagnosis of immune-mediated thrombocytopenia had been made by the referring veterinarian, and the dog had been treated with prednisolone (2 mg/kg orally every 24 h) and vincristine (0.012 mg/kg intravenously once). Three days after the introduction of immunosuppressive therapy, the dog’s clinical conditions further worsened and the murmur features changed. At the time of referral, the dog was markedly depressed and hyperthermic (rectal temperature 39.8 °C). Cardiac auscultation revealed a loud (4/6) left basilar to-and-fro murmur. The heart rate was 56 beats per minute, with a regular rhythm and synchronous and bounding femoral arterial pulses. The rest of the physical examination was within normality. Clinicopathologic abnormalities were limited to severe leukocytosis (white blood cells 55 × 10^9^/L; reference interval 6–17 × 10^9^/L) with neutrophilia and a left shift (neutrophils 45 × 10^9^/L; reference interval 3–12 × 10^9^/L; bands 1.6 × 10^9^/L; reference interval 0–0.5 × 10^9^/L).

A transthoracic echocardiography was then performed using an ultrasound unit (iE33 ultrasound system, Philips Healthcare, Monza, Italy) equipped with phased array transducers (3–8 and 1–5 MHz) and continuous electrocardiographic tracing (Figure 1). Two-dimensional examination showed mild left atrial (left atrial-to aortic ratio 1.7; reference limit < 1.6 [10]) and left ventricular enlargement (Simpson’s method of discs-derived end-diastolic and end-systolic volume indexed to body surface area 112 mL/m^2^ and 19 mL/m^2^, respectively; reference interval 53–93 mL/m^2^ and 24–52 mL/m^2^, respectively [11]), and a hyperdynamic left ventricular systolic function (Teichholz method-derived shorteing fraction 66%; reference interval 30–49% [12]; Simpson’s method of discs-derived ejection fraction 83%; reference interval 48 ± 8 [11]). Additionally, fluttering vegetative lesions (the largest of 5.5 × 10 mm) were evident on all the aortic valve cusps. Color Doppler interrogation of the left ventricular outflow tract revealed a turbulent, antegrade, high-velocity systolic blood flow across the aortic valve (peak velocity 6 m/sec) and a mild diastolic aortic insufficiency (Appendix A). On closer inspection, especially from the right parasternal short-axis view, aortic vegetations seemed to organize into a large (23 × 35 mm), hyperechoic mass-like lesion. The latter extended toward the right-sided cardiac chambers and involved the septal leaflet of the tricuspid valve, which appeared thickened and irregular compared to the parietal one. Moreover, a tangled fistulous tract was visible within the periannular mass-like lesion (Appendix A). Fistulization was further confirmed by color Doppler, which demonstrated a turbulent jet originating from the aortic root flowing into the right ventricle through two distinct openings, one immediately distal to the septal insertion of the tricuspid valve and the other closer to the right ventricular outflow tract (Appendix A). At this level, continuous-wave Doppler analysis documented a high-velocity (3.7 m/s), diastolic, left-to-right shunt. On concurrent electrocardiographic monitoring, third-degree AVB was evident, and mild mitral and tricuspid valve regurgitation was detected associated with non-conducted P waves on color Doppler examination. No other echocardiographic abnormalities were identified. Based on these findings, a diagnosis of aortic valve steno-insufficiency due to IE complicated by ACF and complete AVB was made. The tricuspid valve lesion was primarily considered an extension of the aortic valve infective process, although a thrombus could not be conclusively ruled out. A six-lead electrocardiogram was subsequently obtained for further evaluation of the dog’s cardiac rhythm, confirming the presence of third-degree AVB associated with a monomorphic ventricular escape rhythm (Figure 2). Further diagnostics and therapeutic options were discussed with the owners (including epicardial pacemaker implantation followed by hospitalization in the intensive care unit for proper post-operative management and intravenous antibiotic administration), but these were refused. Due to the dog’s poor clinical condition and infaust prognosis, the owner elected euthanasia and consented to necropsy evaluation.

### 2.2. Gross Examination and Histopathology

Gross evaluation of the heart was made using the inflow–outflow method. A mild subaortic stenosis along with severe and diffuse vegetative lesions covering all the aortic valve cusps was identified. Additionally, a fistulous tract (thickness 4 mm) through the membranous ventricular septum was present (Figure 3). The fistula was engorged with a hematic clot and had a cranio-caudal direction from the aortic valve cusps to the septal tricuspid leaflet. A bulging hematic clot tenaciously adherent to the ventricular face of the tricuspid septal leaflet was evident; moreover, the tricuspid leaflets were irregularly thickened (Figure 4A). Other findings at necropsy included multifocal hepatic, renal, and splenic infarcts and abscesses, and mild pulmonary hemorrhages. Histopathology of aortic and tricuspid valves revealed a severe endocarditis with numerous exophytic aggregates of basophilic, bacillary, Gram-positive bacteria embedded in a fibrin meshwork with numerous degenerated neutrophils and erythrocytes (Figure 4B). Similar findings were evident within the interventricular septum, particularly close to the fistulous tract. Disseminated thrombosis of medium and small-caliber vessels with necrosis of the parenchyma and infiltration of karyolitic neutrophils were observed in the kidneys, spleen, and liver.

## 3. Discussion

Infective endocarditis is a rare disease, but its impact is significant. In dogs, the reported prevalence is 0.09–6.6%, and the median survival time ranges from 3 to 476 days according to the valve infected and the bacteria involved (e.g., the median survival time of aortic IE due to Bartonella and non-Bartonella infections is 3 days and 14 days, respectively, whereas the median survival time of mitral IE is significantly longer) [2]. The rarity and rapid progression of this condition require a high diagnostic awareness to avoid delayed recognition or misdiagnosis. Transthoracic echocardiography represents a diagnostic mainstay given its wide availability, limited invasiveness, and ability to detect the lesions characteristic of IE (i.e., rapidly-evolving irregular-shape vegetations adherent to the aortic and/or mitral valve associated with new regurgitant jets) [1,2,9]. However, due to the dynamic nature of this disease, echocardiographic findings may evolve rapidly, sometimes unpredictably.

This can occur when bacteria excrete lytic enzymes that destroy neighboring tissues, resulting in periannular complications such as abscesses and sinus of Valsalva pseudoaneurysms [13,14,15]. Tissue necrosis associated with infection can promote the rupture of these structures, leading to ACF development [14]. The hemodynamic consequences of this acquired shunt depend on its location and size, as the relative pressures of the chambers communicating with the aorta and the diameter of the rupture influence the direction and magnitude of blood flow [16]. In the case described here, the fistulous tract connected the aortic root and the right ventricle. Since a high-pressure gradient exists between these two compartments both during systole and diastole, a continuous high-velocity, left-to-right shunt would have been expected [16,17]. Nevertheless, color Doppler analysis from the dog in this report demonstrated a left-to-right flow exclusively during diastole, while no shunting flow was identifiable during systole. Such an intriguing finding was likely the result of the transient occlusion of the fistula entrance during ventricular systole by the opening of aortic valve cusps [17,18].

In humans, ACFs occur in less than 2.2% of native aortic valve IE, fistulae originate in similar rates from the three sinuses of Valsalva, and the four cardiac chambers are equally involved in the fistulous tracts [14]. If not promptly diagnosed, human ACFs caused by IE have a severe clinical course, leading to heart failure and in-hospital mortality in approximately 60% and 40% of patients, respectively [14]. In humans, echocardiography is the technique of choice for the diagnosis and localization of IE-related fistulae and the quantification of the resulting intra-cardiac shunts [9,13,14]. The sensitivity of transthoracic and transesophageal echocardiography for the diagnosis of periannular complications of IE is about 50% and 90%, respectively, while more than 90% specificity has been reported for both modalities [8]. Unlike human IE, association of canine IE with intracardiac defects has been described in only four case reports [19,20,21,22], of which just two providing an echocardiographic description [19,21]. In two of these publications, a Gerbode defect (communication between the left ventricle and the right atrium) was identified [20,21]; in another case, a fistula that crossed the interventricular septum developed from a suspected myocardial abscess [22], whereas in the remaining report, a coronary artery-to-right atrial fistula was documented [19].

Infrequently, ACF can act as a route of infection spread allowing bacteria to colonize tissues usually not affected by IE, such as the tricuspid valve. In this extremely unusual circumstance, to date reported only in few human case reports [18,23,24] and once in the dog [19], the septal tricuspid leaflet can undergo the same structural changes as the aortic valve (i.e., valvular thickening, distorting, and closing inappropriately). In the more advanced cases, extensive periannular tissue erosion can also structurally damage the atrioventricular conduction system, which lies directly above the insertion of the septal leaflet of the tricuspid valve [25]. Consequently, AVBs can develop, resulting in further hemodynamic instability and clinical deterioration [13,19,21,22,25,26]. During IE, additional mechanisms of AVBs include development of myocarditis and myocardial abscess leading to perinodal interstitial edema and inflammatory infiltrates [22,27,28], as well as septic coronary embolism leading to ischemic damage of the nodal tissue [29]. To the Authors’ knowledge, the coexistence of aortic and tricuspid valve IE, intracardiac shunt, and complete AVB has been previously documented only in one dog [21]. However, in that case, cardiovascular examination was performed exclusively on the day of referral; thus, it remained unresolved if the intracardiac communication was the result of IE or a concomitant congenital heart defect [21]. In contrast, the dog in this report underwent multiple echocardiographic controls over time, allowing us to rule out cardiovascular abnormalities other than SAS before the occurrence of IE. Accordingly, it is reasonable to assume that our dog first developed aortic valve IE, likely promoted by SAS and the immunosuppressant therapy [1,2]. Then, extension of the infective process with involvement of the neighboring tissues rapidly occurred, leading to the development of ACF and AVB in a few days. In our opinion, ACF represented an anatomical prerequisite for secondary infection of the tricuspid valve.

The left ventricular hyperdynamic state represents an additional intriguing echocardiographic sign from the case described herein. The combination of aortic insufficiency and third-degree AVB was a likely explanation for such finding, since both conditions can increase the left ventricular end-diastolic volume [30,31]. Given the mild insufficiency and the severe bradyarrhythmia, we hypothesized that third-degree AVB represented the predominant factor in our dog. During bradyarrhythmias, an increased end-diastolic left ventricular volume develops as the result of an increase in the duration of the filling time. The increased preload, in turn, enhances the force of contraction according to the Frank–Starling mechanism [30,31]. This allows ejection of a larger stroke volume, with the aim of trying to maintain stable cardiac output despite the decreased heart rate.

A final consideration should be made for possible concurrent cardiovascular diseases which, combined with IE, could have further impaired the hemodynamic stability of our dog. Specifically, arrhythmogenic right ventricular cardiomyopathy (ARVC) could be included in the list of potential differential diagnosis in any Boxer who experiences a cardiac-related death, to which ARVC contributes primarily through severe ventricular tachyarrhythmias and ventricular systolic dysfunction [32]. Nevertheless, several factors made ARVC unlikely in our dog. First, ARVC usually represents an adult-onset myocardial disease and its occurrence before three years is rare [32,33]. Second, typical echocardiographic (e.g., right ventricular enlargement and systolic dysfunction, left ventricular systolic dysfunction [32]) and electrocardiographic (i.e., premature ventricular ectopic complex [32]) signs of ARVC were not identified in cardiological investigations performed before referral to our institution or on the day of IE diagnosis. Third, typical ARVC pathological features (e.g., fibrofatty myocardial infiltration [32]) were lacking in the post-mortem evaluation.

In conclusion, this report describes the exceptionally rare coexistence of multiple periannular complications of aortic valve IE, providing additional evidence for the dynamic and often unpredictable behavior of canine endocarditis. Moreover, this case represents an excellent example of the importance of interpreting echocardiographic findings in light of the patient’s history and physical examination as well as of the disease pathophysiology, especially when facing unusual echocardiographic pictures.

## Figures and Tables

**Figure 1 animals-11-00690-f001:**
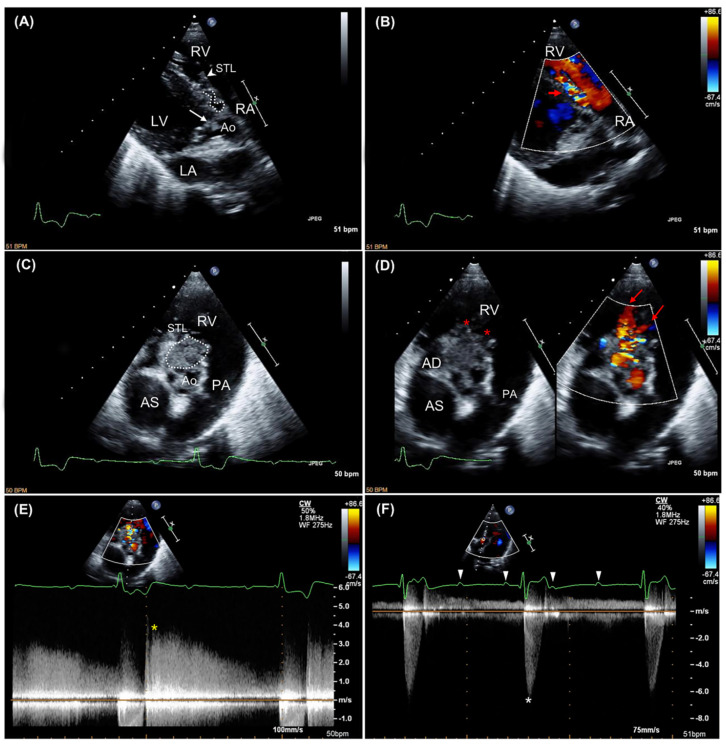
Two-dimensional (**A**) and color Doppler (**B**) findings obtained from an optimized right parasternal long-axis five-chamber view. Aortic valve vegetations (white arrow) and irregular thickening of the septal tricuspid leaflet (white arrowhead) are evident. A fistulous tract opens distal to the septal insertion of the tricuspid valve (white dotted line), leading to a diastolic left-to-right shunt (red arrow). The blood moving from the right atrium to the right ventricle during diastole allows a better appreciation of the temporal phase of the shunt. Two-dimensional (**C**), color (**D**) and continuous-wave Doppler (**E**) findings obtained from an optimized right parasternal short-axis view. Aortic and tricuspid valve thickening are appreciable. Additionally, a hyperechoic mass-like periannular lesion is visible (white dotted circle), characterized by a fistulous tract with two distinct openings (red asterisks) through which two diastolic left-to-right shunting jets occur (red arrows). The yellow asterisk indicates the peak velocity of the shunting flow. Continuous-wave Doppler findings obtained from a left apical five-chamber view (**F**). A high-velocity, antegrade systolic blood flow moves across the aortic valve, consistent with severe stenosis (white asterisk). Line segment with an “X” and green dot on the right side of each two-dimensional image represents the focal zone. On the electrocardiographic tracing, white arrowheads indicate non-conducted P waves, consistent with third-degree atrioventricular block. Ao, aortic valve; LA, left atrium; LV, left ventricle; PA, pulmonary artery; RA, right atrium; RV, right ventricle; STL, septal tricuspid leaflet.

**Figure 2 animals-11-00690-f002:**
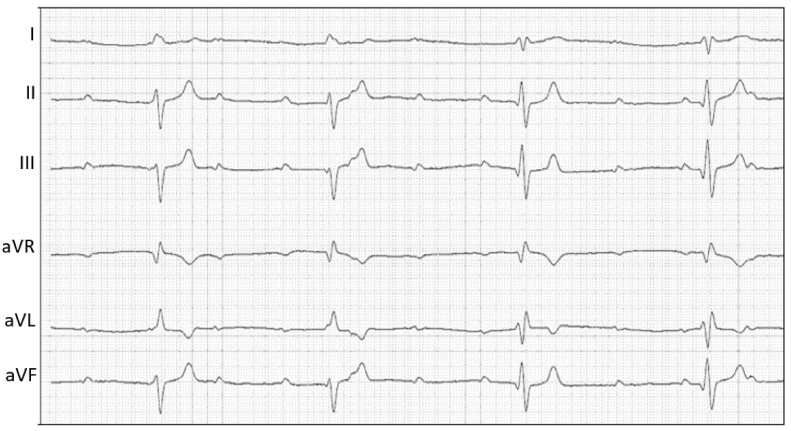
Six-lead electrocardiogram I, II, III, aVR, aVL, aVF. Notice the discrepancy between the atrial rate (approximately 136 beats per minute) and the slow ventricular rhythm (approximately 50 beats per minute). The QRS complexes are wide (duration, 100 milliseconds; reference range, <70 milliseconds) and have a negative polarity in the caudal lead tracings (II, III, and aVF). These findings are indicative of a third-degree atrioventricular block with a monomorphic ventricular escape rhythm. Paper speed = 50 mm/s; 5 mm = 1 mV.

**Figure 3 animals-11-00690-f003:**
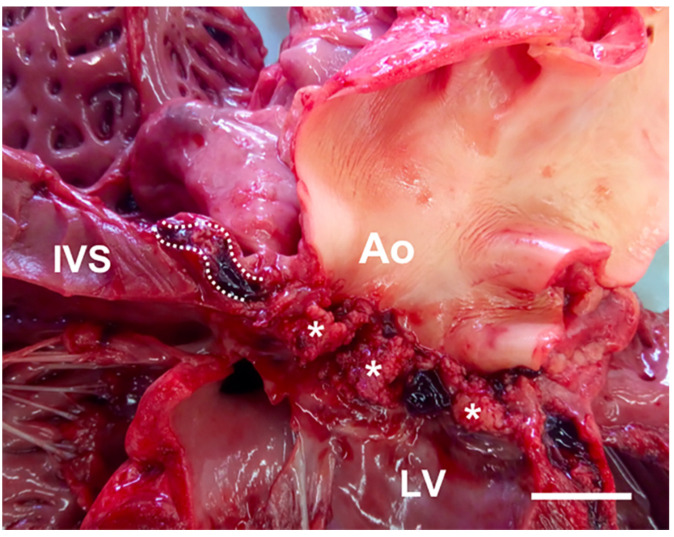
Macroscopic close-up of the left ventricular outflow tract. Vegetative lesions of the aortic valve cusps (white asterisks) are evident. In this view, a fistulous tract is also identifiable (white dotted lines) developing within the membranous ventricular septum. Ao, aortic valve; LV, left ventricle; IVS, interventricular septum. White scale bar = 2 cm.

**Figure 4 animals-11-00690-f004:**
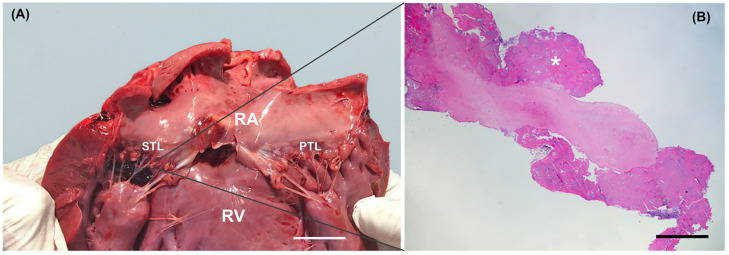
Macroscopic view of the right side of the heart (**A**). Nodular irregularities of the tricuspid leaflets are appreciable. Histopathology of a septal tricuspid valve leaflet. White scale bar = 2 cm. (**B**). On the endocardial surface, exophytic bacterial and fibrin aggregates (white asterisk) are present. Hematoxylin-eosin, 20×. Black scale bar = 500 μm. RA, right atrium; RV, right ventricle; STL, septal tricuspid leaflet. PTL, parietal tricuspid leaflet.

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
