# Peer review of "Unusual Presentation of Aortic Valve Infective Endocarditis in a Dog: Aorto-Cavitary Fistula, Tricuspid Valve Endocarditis, and Third-Degree Atrioventricular Block"

_animals, 2021, doi:10.3390/ani11030690_

Round 1

Reviewer 1 Report

The manuscript is well written and reports a very interesting case with an unusual presentation of aortic endocarditis in a dog.

Please consider the following comments:

INTRODUCTION

Well done!

MATERIALS, METHODS, AND RESULTS

Lines 69-70: Please correct in “due to a previously diagnosed a mild SAS”

Lines 79-80: “moderate (4/6) left basilar to-and-fro murmur”. Usually, a 4/6 murmur grade is considered a “loud” intensity (not “moderate”). Please see: Ljungvall I, et al. Murmur intensity in small-breed dogs with myxomatous mitral valve disease reflects disease severity. J Small Anim Pract 2014;55:545-50. I recommend changing into “loud”, or justify why you reported “moderate”.

Lines 82-83: “Clinicopathologic abnormalities were limited to severe leukocytosis with neutrophilia and a left shift”. Please report the WBC count and specify what do you mean with “left shift”: band neutrophils (n and %)? toxic change in neutrophils? etc… Did you measure cardiac troponin I? This is usually high in dogs with endocarditis showing atrioventricular blocks.

Lines 88-90: “…preserved systolic function. Simpson's method of discs-derived end-diastolic and end-systolic volume indexed to 89 body surface area 112 mL/m2 and 19 mL/m2”. Please add parameters of LV systolic function (LV FS % and/or LV EF%). Based on your data a hyperdynamic LV was probably present (EF around 80%). This is reasonable, nice!! But please explain this interesting hemodynamic finding.

It seems that the author did not record an ECG tracing. Please include an additional figure with a 3-lead or 6-lead standard ECG confirming the 3-AVB.

Gross Examination and Histopathology: Did you find any possible site of infection causing the bacteraemia/endocarditis?

DISCUSSION

Lines 217-221: “To the Author’s knowledge, the coexistence of aortic and tricuspid valve IE, intracardiac shunt and complete AVB has been previously documented only in one dog [19]. However, in that case, cardiovascular examination was performed exclusively on the day of referral; thus, it remained unresolved if the intracar-220 diac communication was the result of IE or a concomitant congenital heart defect [19]”. I would suggest to compare your results with a previous study reporting a similar case (Boxer) with endocarditis, aortic perivalvular abscess with subsequent fistulization, and 3AVB [Vezzosi T, et al. ECG of the Month. Atrioventricular block (AVB). J Am Vet Med Assoc. 2016 May 1;248(9):1004-6].

Please briefly review the etiology of 3AVB in dogs, and expand the discussion regarding the etiology of AVB during aortic endocarditis: myocarditis, coronary thromboembolism, abscess formation, etc. considering the following references: Robertson BT, et al. Complete heart block associated with vegetative endocarditis in a dog. J Am Vet Med Assoc 1972;161:180–184; MacDonald KA, et al. A prospective study of canine infective endocarditis in northern California (1999–2001): emergence of Bartonella as a prevalent etiologic agent. J Vet Intern Med 2004;18:56–64; Sykes JE, et al. Clinicopathologic findings and outcome in dogs with infective endocarditis: 71 cases (1992-2005). J Am Vet Med Assoc 2006;228:1735–1747; Santilli RA, et al. Long-term Intrinsic Rhythm Evaluation in Dogs with Atrioventricular Block. J Vet Intern Med. 2016 Jan-Feb;30(1):58-62.

REFERENCES

Please consider implementing the bibliography with the above recommended previously published studies on the topic of your manuscript.

Author Response

REVIEWER 1

Comments and Suggestions for Authors

The manuscript is well written and reports a very interesting case with an unusual presentation of aortic endocarditis in a dog.

We are grateful to the Reviewer for her/his useful comments and precious indications.

MATERIALS, METHODS, AND RESULTS

Lines 69-70: Please correct in “due to a previously diagnosed a mild SAS”

The sentence has been correct to be grammatically correct.

Lines 79-80: “moderate (4/6) left basilar to-and-fro murmur”. Usually, a 4/6 murmur grade is considered a “loud” intensity (not “moderate”). Please see: Ljungvall I, et al. Murmur intensity in small-breed dogs with myxomatous mitral valve disease reflects disease severity. J Small Anim Pract 2014;55:545-50. I recommend changing into “loud”, or justify why you reported “moderate”.

The definition of murmur intensity has been changed from ‘moderate’ to loud’.

Lines 82-83: “Clinicopathologic abnormalities were limited to severe leukocytosis with neutrophilia and a left shift”. Please report the WBC count and specify what do you mean with “left shift”: band neutrophils (n and %)? toxic change in neutrophils? etc… Did you measure cardiac troponin I? This is usually high in dogs with endocarditis showing atrioventricular blocks.

As required, additional information on the WBC have been reported. Regrettably, cardiac troponin has not been measured due to owner’s financial constraints.

Lines 88-90: “…preserved systolic function. Simpson's method of discs-derived end-diastolic and end-systolic volume indexed to 89 body surface area 112 mL/m2 and 19 mL/m2”. Please add parameters of LV systolic function (LV FS % and/or LV EF%). Based on your data a hyperdynamic LV was probably present (EF around 80%). This is reasonable, nice!! But please explain this interesting hemodynamic finding.

We agree with the Reviewer about the definition of the left ventricular systolic function systolic of our dog, namely hyperdynamic. As indicated, we have added the EF% and FS% values and described the causes of this hemodynamic finding in the discussion.

It seems that the author did not record an ECG tracing. Please include an additional figure with a 3-lead or 6-lead standard ECG confirming the 3-AVB.

According to the Reviewer’s request, a Figure (Figure 2) with a 6-lead ECG tracing has been provided.

Gross Examination and Histopathology: Did you find any possible site of infection causing the bacteraemia/endocarditis?

Unfortunately, no conclusive original sites of infection have been identified. Since the inability/impossibility to detect an obvious site of infection represents a frequent issue in animals with infective endocarditis, a sentence has been added in the introduction to make explicit this to the readers.

DISCUSSION

Lines 217-221: “To the Author’s knowledge, the coexistence of aortic and tricuspid valve IE, intracardiac shunt and complete AVB has been previously documented only in one dog [19]. However, in that case, cardiovascular examination was performed exclusively on the day of referral; thus, it remained unresolved if the intracar-220 diac communication was the result of IE or a concomitant congenital heart defect [19]”. I would suggest to compare your results with a previous study reporting a similar case (Boxer) with endocarditis, aortic perivalvular abscess with subsequent fistulization, and 3AVB [Vezzosi T, et al. ECG of the Month. Atrioventricular block (AVB). J Am Vet Med Assoc. 2016 May 1;248(9):1004-6].

We thank the reviewer for the interesting reference provided; we have integrated it in our discussion as well as in the reference list. Concerning the comparison of the dog from our report with other cases described in literature, we initially decided to compare our case with the more similar one available in canine literature (i.e., the only previous report [Peddle et al., J Vet Cardiol 2008] which demonstrated, as we also did, the rare ‘triad’ of complications, namely 1) intracardiac defect, 2) tricuspid valve endocarditis and 3) third-degree atrioventricular block). We still believe that it would be preferable to compare our dog with the case by Peddle et al. rather than the case by Vezzosi et al. because: 1) the case report from Vezzosi et al. did provide neither echocardiographic figures/videos nor pathological images to confirm the clinical suspect and to allow detailed comparison with our case; in contrast, Peddle et al. provided either echocardiographic and pathologic images; 2) the case from Vezzosi et al. had no infective involvement of tricuspid valve; in contrast, Peddle et al. demonstrated tricuspid valve endocarditis (as in the present report).  

Please briefly review the etiology of 3AVB in dogs, and expand the discussion regarding the etiology of AVB during aortic endocarditis: myocarditis, coronary thromboembolism, abscess formation, etc. considering the following references: Robertson BT, et al. Complete heart block associated with vegetative endocarditis in a dog. J Am Vet Med Assoc 1972;161:180–184; MacDonald KA, et al. A prospective study of canine infective endocarditis in northern California (1999–2001): emergence of Bartonella as a prevalent etiologic agent. J Vet Intern Med 2004;18:56–64; Sykes JE, et al. Clinicopathologic findings and outcome in dogs with infective endocarditis: 71 cases (1992-2005). J Am Vet Med Assoc 2006;228:1735–1747; Santilli RA, et al. Long-term Intrinsic Rhythm Evaluation in Dogs with Atrioventricular Block. J Vet Intern Med. 2016 Jan-Feb;30(1):58-62.

Following the suggestion of the Reviewer, we have expanded the description of mechanisms contributing to the development of atrioventricular blocks in patients with infective endocarditis, and have integrated the reference list with pertinent references.

REFERENCES

Please consider implementing the bibliography with the above recommended previously published studies on the topic of your manuscript.

The reference list has been significantly implemented by the addition of several new publications, including many of those proposed by the Reviewer and others pertinent to the new sentences that we have added.

Reviewer 2 Report

This is a nice case report with interesting observations. Labeling of images could be a bit larger. 

No major concerns. Some minor English edits would be useful.

Author Response

REVIEWER 2

This is a nice case report with interesting observations. Labeling of images could be a bit larger.

We thank the Reviewer for his/her positive judgment. As required, the labelling of images has been increased.

Reviewer 3 Report

The case report “Unusual presentation of aortic valve infective endocarditis in a dog: aorto-cavitary fistula, tricuspid valve endocarditis and third-degree atrioventricular block” written by Romito et al. is an interesting, but tragic case study of a pet boxer who had to be euthanized.  In its current form it is unacceptable for publication.

Major Comment: 

The most obvious unanswered question is what caused the infective endocarditis.  As the authors correctly point out, there are several predisposing factors.  However, the one factor they did not point out was arrhythmogenic right ventricular cardiomyopathy.  It is so prevalent in Boxers; it is even referred to as Boxer cardiomyopathy.  It also has incomplete penetrance and in humans, has been associated with both infective endocarditis and subaortic stenosis.  Having studied the molecular mechanisms of cardiomyopathy, I have seen changes in excitation-contraction before seeing hypertrophy.  Consequently, the young age of the animal and lack of overt hypertrophy does not rule this out as a possibility.  Therefore, if possible, this reviewer would like the authors to at least look for a striatin deletion, preferably more.

Nevertheless, this reviewer will acknowledge it may be impossible at this time.  If not possible, please include a couple paragraphs explaining how it might be a possibility. 

In conclusion, the manuscript was beautifully written and quite interesting.  I look forward to the revision.

Minor Comments:

Please include scale bars in figures 2 and 3.

If possible, include a higher resolution/higher magnification for figure 2B.   

Author Response

REVIEWER 3

Comments and Suggestions for Authors

The case report “Unusual presentation of aortic valve infective endocarditis in a dog: aorto-cavitary fistula, tricuspid valve endocarditis and third-degree atrioventricular block” written by Romito et al. is an interesting, but tragic case study of a pet boxer who had to be euthanized.  In its current form it is unacceptable for publication.

We are grateful to the Reviewer for his/her interesting suggestions aimed to improve the quality of the manuscript. Concerning the outcome of the case, regrettably it was fatal, but it is important to underline that this was related to the underlying, coexisting conditions rather than to a delayed/erroneous diagnosis or improper management/questionable clinical choices.

Aortic endocarditis is usually associated to a dramatically short survival. Specifically, the survival of this condition appears to be significantly shorter than the endocarditis of the mitral valve (according to MacDonald et al., J Vet Inter Med 2004;18:56-64 the median survival time of aortic endocarditis due to Bartonella and non-Bartonella infections was only 3 days and 14 days, respectively, whereas the median survival time in the case of the mitral involvement was 540 days). This additional information has been added in the text.

To further worsen the clinical picture, it should be considered the concurrent third-degree atrioventricular block. The survival of dogs with this bradyarrhythmia is positively associated with the implantation of a pacemaker, and the quality of life and survival time of dogs with third-degree AVB dramatically decrease is such procedure is not performed (Schrope and Kelch. J Am Vet Med Assoc 2006;228:1710-7). Unlikely, the association of infective endocarditis and third-degree AVB represent an extremely challenging condition, as the ongoing infective disease process represents, actually, a well-established contraindication for the conventional procedure of pacemaker application (i.e., the minimally-invasive, catheter-based endovascular one) (Orton J Vet Cardiol 2019;22:65-71). The only way to try to manage a such tremendous association of concomitant life-threatening conditions is the surgical application of an epicardial pacemaker (Orton J Vet Cardiol 2019;22:65-71), followed by hospitalization in the intensive care unit for a medium-/long-term administration of intravenous antibiotics. This possibility was discussed with the owner but declined. To clarify this, it has been specified in the case description.

Lastly, it should be considered that our dog developed also an acquired intracardiac fistula, a condition that, if not promptly treated surgically under cardiopulmonary bypass (the latter a procedure that, to date, is limited to few specialized veterinary centres around the world and typically recommended only to dog with a stable clinical condition due to the high peri- and post-procedural complication and mortality rate), has a severe clinical course in humans (i.e., heart failure and in-hospital mortality in approximately 60% and 40% of patients, respectively). Given the above, the final choice of performing euthanasia did not represent an empirical short-cut but rather an evidence-based, although sad, choice.   

Major Comment:

The most obvious unanswered question is what caused the infective endocarditis.  As the authors correctly point out, there are several predisposing factors.  However, the one factor they did not point out was arrhythmogenic right ventricular cardiomyopathy.  It is so prevalent in Boxers; it is even referred to as Boxer cardiomyopathy.  It also has incomplete penetrance and in humans, has been associated with both infective endocarditis and subaortic stenosis.  Having studied the molecular mechanisms of cardiomyopathy, I have seen changes in excitation-contraction before seeing hypertrophy.  Consequently, the young age of the animal and lack of overt hypertrophy does not rule this out as a possibility.  Therefore, if possible, this reviewer would like the authors to at least look for a striatin deletion, preferably more. Nevertheless, this reviewer will acknowledge it may be impossible at this time.  If not possible, please include a couple paragraphs explaining how it might be a possibility.

We agree with the Reviewer about the point that the most obvious unanswered question is what originally caused the infective endocarditis. At regard, it is important to underline that sometimes it is not possible to understand the location of the original site of infection, even in cases subjected to an extensive diagnostic work up. Since this problem/limitation is not that rare in small animal practice, a sentence on this regard has been added in the introduction of the manuscript (associated with pertinent references), to make clear this point. At the same time, such knowledge could be useful for readers to understand that the lack of identification of the original site of infection neither represents a criterion to rule out endocarditis nor makes more fragile a diagnosis of endocarditis.

Concerning the possible link between infective endocarditis and arrhythmogenic right ventricular cardiomyopathy (ARVC), both in humans and dogs ARVC does not represent a predisposing factor for endocarditis; accordingly, we did not include ARVC among the list of well-accepted/well-recognized predisposing factors from canine literature. Nevertheless, we agree with the idea that ARVC should be always included in the list of potential/theoretical differential diagnosis in any Boxer who experience a cardiac-related death. Moreover, we agree about the possibility that this cardiomyopathy could manifest concurrently with additional cardiovascular disorders. This is likely to be due to the relatively high prevalence of ARVC in a breed (the Boxer) which is highly predisposed also to other cardiovascular diseases (e.g., subaortic stenosis, pulmonic stenosis), rather than to be the result of a mutation shared between multiple disease entities. Nevertheless, several features from the case described herein made, in our opinion, unlikely the presence of an ARVC concomitant with infective endocarditis. First, the echocardiographic and electrocardiographic features detected on the day of referral to our institution were not typical for ARVC (unremarkable right ventricular morphology and function; hyperdynamic left ventricular function). Second, post-mortem evaluation did not identify typical pathological features of ARVC. Third, it should be considered that the dog from this report underwent multiple cardiologic evaluations before referral to our institution (mainly with the aim to monitor the subaortic stenosis), and that none of the previous investigations identified any clinical or diagnostic sign potentially attributable to ARVC (e.g., no episodes of syncope; no abnormalities of the right as well as left ventricular structure and function; no ventricular ectopic complexes). Fourth, Boxer ARVC is usually an adult-onset myocardial disease, which occur rarely in dogs ≤3 years.

Theoretically, the execution of a test to look for striatin mutation could have provided further information on ARVC but at this time, regrettably, it is not possible to be perform. However, in the light of points above listed, the utility of that test in the present case appears questionable. Such questionability is also based on the fact that even if the dog would have tested positive for the mutation (in that case, an heterozygous state would have been expected since the heterozygous state could be detected even in dogs without the typical ARVC/DCM phenotype, contrary to the homozygous state which is detected in dogs with an overt disease’s phenotype), it would have been a dog with an ‘occult’ ARVC since neither clinical signs nor echocardiographic and electrocardiographic findings related to the cardiomyopathy were developed. Therefore, in that case, ARVC would have been interpreted as a ‘silent’ comorbidity unrelated to the ongoing clinical disease (i.e., an ‘incidentaloma’); indeed, it is important to underline the cardiovascular compromise developed only once the endocarditis occurred and was the result of its complications, especially the third-degree atrioventricular block.

As request, a section concerning ARVC and the reasons that made this differential unlikely has been included in the discussion.

Minor Comments:

Please include scale bars in figures 2 and 3.

Scale bars in figures 2 and 3 were included.

If possible, include a higher resolution/higher magnification for figure 2B.  

Higher resolution for figure 4B (in the current version of the manuscript) was provided.
